**mSystems**®

# Multiple T6SSs, Mobile Auxiliary Modules, and Effectors Revealed in a Systematic Analysis of the *Vibrio parahaemolyticus* Pan-Genome

Biswanath Jana,[a] Kinga Keppel,[a] Chaya Mushka Fridman,[a] Eran Bosis,[b] Dor Salomon[a]

[a]Department of Clinical Microbiology and Immunology, Sackler Faculty of Medicine, Tel Aviv University, Tel Aviv, Israel
[b]Department of Biotechnology Engineering, Braude College of Engineering, Karmiel, Israel

**ABSTRACT** Type VI secretion systems (T6SSs) play a major role in interbacterial competition and in bacterial interactions with eukaryotic cells. The distribution of T6SSs and the effectors they secrete vary between strains of the same bacterial species. Therefore, a pan-genome investigation is required to better understand the T6SS potential of a bacterial species of interest. Here, we performed a comprehensive, systematic analysis of T6SS gene clusters and auxiliary modules found in the pan-genome of *Vibrio parahaemolyticus*, an emerging pathogen widespread in marine environments. We identified 4 different T6SS gene clusters within genomes of this species; two systems appear to be ancient and widespread, whereas the other 2 systems are rare and appear to have been more recently acquired via horizontal gene transfer. In addition, we identified diverse T6SS auxiliary modules containing putative effectors with either known or predicted toxin domains. Many auxiliary modules are possibly horizontally shared between *V. parahaemolyticus* genomes, since they are flanked by DNA mobility genes. We further investigated a DUF4225-containing protein encoded on an Hcp auxiliary module, and we showed that it is an antibacterial T6SS effector that exerts its toxicity in the bacterial periplasm, leading to cell lysis. Computational analyses of DUF4225 revealed a widespread toxin domain associated with various toxin delivery systems. Taken together, our findings reveal a diverse repertoire of T6SSs and auxiliary modules in the *V. parahaemolyticus* pan-genome, as well as novel T6SS effectors and toxin domains that can play a major role in the interactions of this species with other cells.

**IMPORTANCE** Gram-negative bacteria employ toxin delivery systems to mediate their interactions with neighboring cells. *Vibrio parahaemolyticus*, an emerging pathogen of humans and marine animals, was shown to deploy antibacterial toxins into competing bacteria via the type VI secretion system (T6SS). Here, we analyzed 1,727 *V. parahaemolyticus* genomes and revealed the pan-genome T6SS repertoire of this species, including the T6SS gene clusters, horizontally shared auxiliary modules, and toxins. We also identified a role for a previously uncharacterized domain, DUF4225, as a widespread antibacterial toxin associated with diverse toxin delivery systems.

**KEYWORDS** T6SS, antibacterial, competition, secretion, *Vibrio*

Address correspondence to Dor Salomon, dorsalomon@mail.tau.ac.il, or Eran Bosis, bosis@braude.ac.il.

The authors declare no conflict of interest.

During competition, bacteria deliver toxic cocktails of effectors using specialized, contact-dependent protein secretion systems (1). Gram-negative bacteria often employ the type VI secretion system (T6SS) to gain a competitive advantage over their rivals (2). This system comprises 13–14 core components, in addition to accessory components that differ between systems and may play different regulatory roles (3–6). The structural and regulatory components of T6SS are encoded within large gene clusters that also often encode effectors; effectors are also encoded within auxiliary modules

containing T6SS core components (7–10), or within orphan cassettes (11, 12). The effectors are loaded onto a missile-like structure comprising a tube consisting of stacked rings of hemolysin-coregulated protein (Hcp) hexamers; the tube is capped by a spike complex composed of a valine-glycine repeat protein G (VgrG) trimer sharpened by a proline-alanine-alanine-arginine (PAAR) repeat-containing protein (hereafter referred to as PAAR) (13). The loaded missile is propelled outside of the cell by a contracting sheath that engulfs it (14); the contraction provides sufficient force to penetrate a neighboring cell and deploy the effectors (5).

Many T6SS effector families that contain toxin domains mediating antibacterial activities have been reported in various bacteria. Effectors often target conserved bacterial components in the cytoplasm or periplasm, such as nucleic acids (nucleases) (7, 15–19), the membrane (phospholipases and pore-forming toxins) (10, 11, 20–22), or the peptidoglycan layer (amidases, glycoside hydrolases, and carboxy- and transpeptidases) (12, 23, 24). Additional activities mediated by T6SS effectors include the following: altering the energy balance by hydrolyzing NAD(P)$^+$ (25, 26), ADP-ribosylating the conserved protein FtsZ to inhibit cell division (27), ADP-ribosylating the 23S rRNA to inhibit translation (28), targeting the transamidosome to inhibit protein synthesis (29), deaminating the target cell's DNA (30), and synthesizing the toxic molecule (p)ppApp (31). Antibacterial T6SS effectors are encoded adjacent to cognate immunity proteins that protect against self- or kin-intoxication (2, 32). Several experimental and computational approaches have been used to identify effector and immunity (E/I) pairs (2, 7, 11, 12, 29, 33–36). Nevertheless, because T6SS effectors employ diverse mechanisms for secretion (13, 37–39), and therefore lack a canonical secretion signal, it is estimated that many more effectors still await discovery.

*Vibrio parahaemolyticus* is an emerging pathogen that inhabits marine and estuarine environments (40). Previous works revealed that all investigated *V. parahaemolyticus* isolates contain a T6SS on chromosome 2, named T6SS2, whereas pathogenic isolates encode another T6SS on chromosome 1, named T6SS1 (41–43). The presence of additional T6SSs in this species remains to be investigated. T6SS2 was recently shown to mediate antibacterial activities; however, its effector repertoire remains unknown (11, 44). T6SS1, which has been studied in several isolates (7, 11, 36, 42, 43), also mediates antibacterial activities. T6SS1 deploys 2 conserved antibacterial E/I pairs, which are encoded by the main T6SS1 gene cluster, as well as "accessory" E/I pairs that differ between isolates and diversify the effector repertoire (36, 45). To date, few accessory T6SS1 E/I pairs were found in auxiliary T6SS modules containing a gene encoding VgrG (7), or as orphan operons that often reside next to DNA mobility elements (11, 36, 42). Only a handful of these accessory T6SS1 E/I pairs have been experimentally validated (7, 11, 36). Notably, since most vibrios are naturally competent (46, 47), horizontal gene transfer (HGT) may play a role in the acquisition and evolution of the T6SS E/I pair repertoire (48, 49).

In this study, we sought to reveal the collective repertoire of T6SS gene clusters and auxiliary modules in *V. parahaemolyticus*, as well as to identify new effectors. By systematically analyzing 1,727 *V. parahaemolyticus* genomes, we identified 4 types of T6SS gene clusters and many distinct, widespread auxiliary modules predicted to encode diverse effectors; the vast majority of the auxiliary modules, as well as 2 of the T6SS gene clusters, are found next to DNA mobility genes, suggesting that they were acquired via HGT. Intriguingly, most *hcp*-containing auxiliary modules encode a previously undescribed effector with a C-terminal domain of unknown function 4225 (DUF4225). We experimentally showed that this effector is toxic upon delivery to the bacterial periplasm, where it leads to cell lysis. We also identified a downstream-encoded cognate immunity protein that antagonizes the effector's toxic effect. Surprisingly, although several strains of marine bacteria were intoxicated by this effector during competition, others, including 2 *V. parahaemolyticus* strains that do not contain homologs of the cognate immunity protein, were resistant to the attack. Further

analysis revealed that DUF4225 is a widespread toxin domain that is present in polymorphic toxins associated with several protein secretion systems.

## RESULTS

**Four T6SS gene clusters are found in the *V. parahaemolyticus* pan-genome.** We first set out to identify the T6SS gene clusters found in the *V. parahaemolyticus* pan-genome. To this end, we retrieved the sequences of the conserved T6SS core component, TssB (3), from 1,727 available RefSeq *V. parahaemolyticus* genomes (Data set S1). Analysis of the phylogenetic distribution of TssB revealed 4 groups (Fig. 1A) corresponding to 4 distinct T6SS gene clusters in *V. parahaemolyticus* genomes (Fig. 1B and Data set S1 and S2).

T6SS1 and T6SS2 were previously investigated and found to mediate interbacterial competition (7, 11, 36, 43, 44). In agreement with previous analyses on smaller genome data sets (7, 11, 43), we identified T6SS2 in nearly all *V. parahaemolyticus* genomes (99%), whereas T6SS1 was identified in 68.3% of the genomes (Fig. 1B and Fig. 2, and Data set S1 and S2). We did not identify known or potential effectors encoded within the T6SS2 gene cluster of the *V. parahaemolyticus* reference strain RIMD 2210633, nor in T6SS2 gene clusters of several other strains, which were manually assessed; however, the T6SS1 clusters contain 2 antibacterial effectors, corresponding to VP_RS06755 (VP1390) and VP_RS06875 (VP1415) in the reference strain RIMD 2210633 (36, 37) (Table 1). Interestingly, we observed some diversity at the end of T6SS1 gene clusters; we found what appears to be duplications (between 1 and 6 copies) of the PAAR-containing specialized effector, corresponding to VP_RS06875 in the reference strain RIMD 2210633 (36).

Two additional T6SS gene clusters, which we named T6SS3 and T6SS4 (Fig. 1B), have a limited distribution in *V. parahaemolyticus* genomes (0.8% and 1.8%, respectively) (Fig. 2 and Data set S1 and S2). T6SS3 is similar to the previously reported T6SS3 of *V. proteolyticus*, which was suggested to have anti-eukaryotic activity and induce inflammasome-mediated cell death in macrophages (50, 51). T6SS4 could be further divided into 3 sub-groups, a to c, with minor differences in gene sequence and content (Fig. 1B and Data set S2). We identified predicted effectors encoded in both the T6SS3 and T6SS4 gene clusters (Fig. 1B and Table 1). Notably, T6SS3 and T6SS4, which were not previously described in *V. parahaemolyticus*, are flanked by transposases and other DNA mobility elements (Fig. 1), suggesting that they have been acquired via HGT. Taken together, these results reveal that the *V. parahaemolyticus* pan-genome contains 2 widespread T6SSs and two T6SSs with limited distribution.

**Diverse and widespread T6SS auxiliary modules contain effectors.** Although T6SS effectors are often encoded within the main T6SS gene clusters, auxiliary modules containing at least 1 secreted T6SS component (i.e., Hcp, VgrG, or PAAR) and downstream-encoded effectors are also common (7–10). Therefore, to identify putative auxiliary T6SS modules, we searched the *V. parahaemolyticus* pan-genome for Hcp, VgrG, and PAAR encoded outside the four main T6SS gene clusters described above. We found diverse putative auxiliary module types widely distributed among the different *V. parahaemolyticus* genomes (Fig. 2, Fig. S1, and Data set S1 and S3). These modules are predominantly found next to DNA mobility genes, such as integrases, recombinases, transposases, phage proteins, or plasmid mobility elements. Notably, some modules contain more than one predicted hallmark secreted T6SS component, and some genomes harbor multiple module types (up to 7 modules in one genome).

PAAR proteins encoded within auxiliary modules are often specialized effectors containing known (e.g., AHH, Ntox15, and NUC nucleases) or predicted C-terminal toxin domains (Table 1), followed by a downstream gene that possibly encodes a cognate immunity protein (Fig. S1 and Data set S3). Homologous PAAR proteins are also encoded within similar auxiliary module configurations in which the toxin domain is encoded by a separate gene as a predicted cargo effector (Table 1 and Data set S3). In contrast, we did not identify auxiliary VgrG proteins containing C-terminal toxin domains; instead, auxiliary VgrG modules carry predicted cargo effectors, some with known activities (e.g., PoNe DNase, NucA/B nuclease, and Lip2 lipase). These

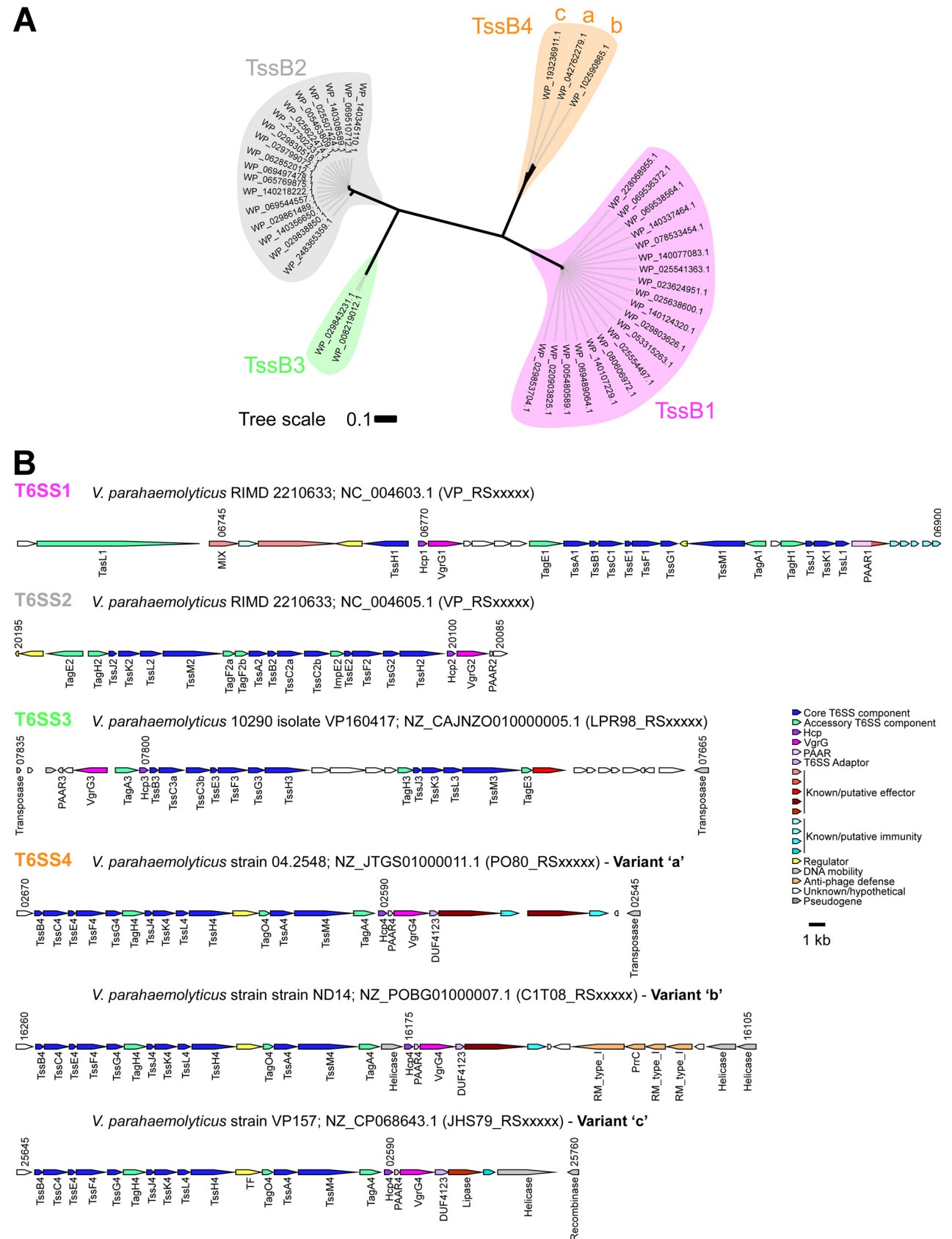

**FIG 1** The *V. parahaemolyticus* pan-genome harbors four T6SSs. (A) Phylogenetic distribution of the T6SS core component TssB encoded within *V. parahaemolyticus* genomes. The evolutionary history was inferred using the neighbor-joining method. The phylogenetic tree was drawn to scale, with

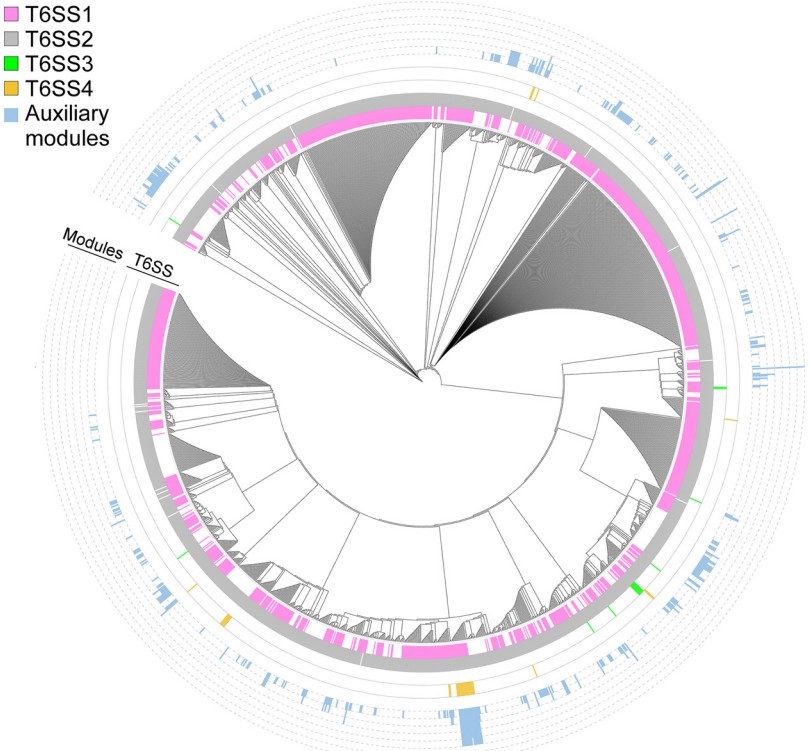

**FIG 2** Distribution of T6SS gene clusters and auxiliary modules in *V. parahaemolyticus*. The phylogenetic tree was based on DNA sequences of *rpoB* coding for DNA-directed RNA polymerase subunit beta. The evolutionary history was inferred using the neighbor-joining method. The height of the blue bars denotes the number of T6SS auxiliary modules per genome.

predicted cargo effectors are mostly encoded downstream of a gene encoding a T6SS adaptor protein, such as DUF4123, DUF2169, and DUF1795 (39, 52) (Fig. S1 and Data set S3).

Hcp-encoding auxiliary modules contain a few, previously uncharacterized, predicted cargo effectors of unknown function (Table 1, Fig. S1, and Data set S3). Interestingly, in most Hcp-containing auxiliary modules that we found, which are located in diverse syntenies, a DUF4225-containing protein is encoded immediately downstream of *hcp* (Fig. S2). Some DUF4225-encoding genes have an adjacent, small gene encoding a protein predicted to contain transmembrane helices (according to a Phobius server analysis [53]) (Data set S3). Based on these findings, we hypothesized that these DUF4225-encoding genes and their downstream adjacent genes are antibacterial T6SS E/I pairs.

**Constructing an "effectorless" surrogate T6SS platform.** A surrogate T6SS platform can be used as a tool to study putative E/I pairs. The major advantage of the surrogate platform is that it only requires constructing a single plasmid to express the putative E/I pair in question (11). Since we have previously reported that the T6SS1 of POR1, a *V. parahaemolyticus* strain RIMD 2210633 derivative, can be used as a surrogate platform to deliver and investigate effectors and modules belonging to T6SS1 from other *V. parahaemolyticus* strains (11), we decided to employ this strategy to investigate the predicted DUF4225-containing effector.

**FIG 1** Legend (Continued)
branch lengths in the same units as those of the evolutionary distances used to infer the phylogenetic tree. The evolutionary distances were computed using the Poisson correction method; they are in the units of the number of amino acid substitutions per site. The letters a, b, and c denote the T6SS4 variant with which the *tssB4* is associated. (B) Representative T6SS gene clusters found in *V. parahaemolyticus* genomes. The strain names, the GenBank accession numbers, and the locus tag annotation patterns are provided. Genes are denoted by arrows indicating the direction of transcription. Locus tags are denoted above, and the names of encoded proteins or domains are denoted below.

**TABLE 1** Predicted effectors in *V. parahaemolyticus* T6SS clusters and auxiliary modules

| Example accession no. | Example gene locus | Found in T6SS or module type | Effector type | Predicted toxin domain | Predicted activity | Homologs in other polymorphic toxin classes | Ref. |
|---|---|---|---|---|---|---|---|
| WP_005480617.1 | VP_RS06755 (VP1390) | T6SS1 | Cargo | Unknown | Cell lysis | No | 37 |
| WP_005493834.1 | VP_RS06875 (VP1415) | T6SS1, PAAR | Specialized/Cargo | AHH | Nuclease | Yes | 36, 61 |
| WP_029843206.1 | LPR98_RS07710 | T6SS3 | Cargo | Unknown | Unknown | Yes | |
| WP_042762256.1 | PO80_RS02570 | T6SS4a,b | Cargo | Unknown | Unknown | No | |
| WP_193237005.1 | JHS79_RS25745 | T6SS4c, Hcp+VgrG | Cargo | Lip2 | Lipase | Yes | |
| WP_065771704.1 | AKH09_RS09365 | PAAR+VgrG+Hcp | Cargo | Unknown | Unknown | Yes | |
| WP_238790300.1 | K6U37_RS14065 | PAAR+VgrG | Cargo | NucA/B | Nuclease | Yes | 61 |
| WP_020841305.1 | H9J98_RS02420 | PAAR | Specialized/Cargo | Ntox15 | Nuclease | Yes | 15, 61 |
| WP_083135234.1 | GPY55_RS17385 | PAAR | Specialized/Cargo | NUC | Nuclease | Yes | 61 |
| WP_102591288.1 | C1T08_RS26340 | PAAR | Specialized | Unknown | Unknown | Yes | |
| WP_102591220.1 | C1S85_RS24675 | PAAR | Specialized | Tme | Membrane-disrupting | Yes | 11 |
| WP_102591225.1 | C1S85_RS24700 | PAAR | Specialized | Unknown | Unknown | Yes | |
| WP_238789479.1 | K6U37_RS04455 | PAAR | Specialized (truncated) | Unknown | Unknown | Yes | |
| WP_129147717.1 | EGL73_RS17180 | VgrG | Cargo | Unknown | Unknown | No | 78 |
| WP_029857615.1 | B5C30_RS14465 | VgrG | Cargo | PoNe | DNase | Yes | 7 |
| WP_238790289.1 | K6U37_RS13990 | VgrG | Cargo | Unknown | Unknown | Yes | 45 |
| WP_086585359.1 | JHS88_RS14235 | Hcp | Cargo | DUF4225 | Cell lysis | Yes | This work |
| WP_195433156.1 | K6U37_RS12660 | Hcp | Cargo | Unknown | Unknown | Yes | |
| WP_228085946.1 | IB292_RS21975 | Hcp | Cargo | Unknown | Unknown | Yes | |

A drawback of our previously reported surrogate platform was the presence of the endogenous T6SS1 effectors of the RIMD 2210633 strain (36), which prevented the use of a possibly sensitive prey strain and thus required the use of a RIMD 2210633-derived strain containing the cognate immunity proteins as prey during competition assays. To enable the use of a surrogate platform during competition against diverse prey strains, we set out to construct an "effectorless" version. To this end, we deleted the genes encoding the reported effector VPA1263 (36) and the co-effector VP1388 (37), and we replaced 2 residues in the predicted active site of the specialized effector VP1415 (36) with alanine, as previously reported (54). In addition, we deleted *vp1133*, which encodes a Histone-like nucleoid-structuring protein (H-NS) that represses T6SS1 activity in *V. parahaemolyticus* (55), to constitutively activate T6SS1 in the surrogate strain. The resulting platform, which we named VpT6SS1$^{Surrogate}$, is active at 30°C in media containing 3% (wt/vol) NaCl, as evident by the expression and secretion of the hallmark VgrG1 protein in a T6SS1-dependent manner (Fig. 3A). Furthermore, VpT6SS1$^{Surrogate}$ enables interbacterial killing of susceptible *V. natriegens* prey during competition mediated by a plasmid-borne VgrG1b auxiliary module belonging to T6SS1 of *V. parahaemolyticus* strain 12–297/B, containing the PoNe DNase effector (7) (Fig. 3B).

**A DUF4225-containing protein is a T6SS1 antibacterial effector.** To determine whether DUF4225-encoding genes and their adjacent downstream genes are antibacterial T6SS E/I pairs, we chose to investigate an Hcp auxiliary module from *V. parahaemolyticus* strain CFSAN018764 (Fig. S2; RefSeq sequence NZ_LHBG01000025.1) encoding an Hcp and a DUF4225-containing protein, hereafter referred to as DUF4225$^{18764}$ (accession numbers WP_065788327.1 and WP_065788326.1, respectively). This strain harbors both T6SS1 and T6SS2 (Data set S1). Since the module's Hcp is more similar to Hcp1 than to Hcp2 (Fig. S3), we reasoned that this Hcp auxiliary module is probably associated with T6SS1; we therefore named this Hcp as Hcp1b.

The Hcp1b module lacked an annotated gene immediately downstream of the gene encoding the predicted effector DUF4225$^{18764}$. Nevertheless, when we performed a manual analysis of the nucleotide sequence, we identified an open reading frame immediately downstream of the predicted effector (positions 296 to 27 in NZ_LHBG01000025.1) (Fig. S2). The identified gene is predicted to encode an 89 amino acid-long protein containing 3 transmembrane helices. Therefore, we hypothesized that this gene encodes a cognate DUF4225$^{18764}$ immunity protein, and we named it Imm4225$^{18764}$.

Using VpT6SS1$^{Surrogate}$, we next set out to investigate the ability of the Hcp1b auxiliary module from strain CFSAN018764 to mediate T6SS1-dependent competition. As shown in Fig. 3C, an arabinose-inducible plasmid encoding the three proteins of the Hcp1b module (pModule; i.e., Hcp1b, DUF4225$^{18764}$, and Imm4225$^{18764}$), but not Hcp1b alone (pHcp1b), mediated the T6SS1-dependent intoxication of *V. natriegens* prey, as indicated by the decline in prey viability when the module was expressed in the parental attacker but not when it was expressed in the T6SS1-inactive mutant (Δ*hcp1*). Expression of Imm4225$^{18764}$ from a plasmid (pImm) rescued the *V. natriegens* prey strain from this attack. Taken together, these results indicate that the Hcp1b auxiliary module of *V. parahaemolyticus* strain CFSAN018764 carries a T6SS1 antibacterial E/I pair. In addition, they suggest that Hcp1b is unable to support T6SS-mediated delivery of the effector in the absence of the canonical T6SS1 Hcp, Hcp1.

**A DUF4225-containing effector leads to cell lysis upon delivery to the periplasm.** Next, we investigated the toxicity mediated by DUF4225$^{18764}$. The arabinose-inducible expression of the effector in the periplasm of *Escherichia. coli* (by fusion to an N-terminal PelB signal peptide), but not in the cytoplasm, led to a clear reduction in the optical density (OD$_{600}$) of the bacterial culture over time (Fig. 4A); the phenotype was similar to the lytic effect of the amidase effector Tse1 from *Pseudomonas aeruginosa* (56), and dissimilar to the effect of the membrane-disrupting effector Tme1 from *V. parahaemolyticus* (11). Lysis was also observed when *E. coli* cells expressing the periplasmic version of DUF4225$^{18764}$ were visualized under a fluorescence microscope. Approximately 90 min after inducing DUF4225$^{18764}$ expression, cells began to shrink and appeared to have lost their cytoplasmic content. Concomitantly, the leakage of

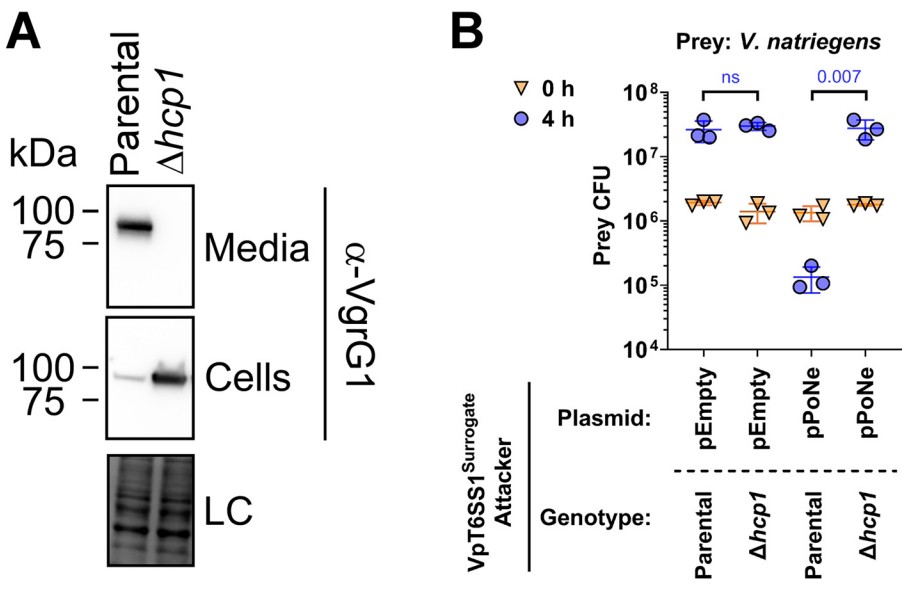

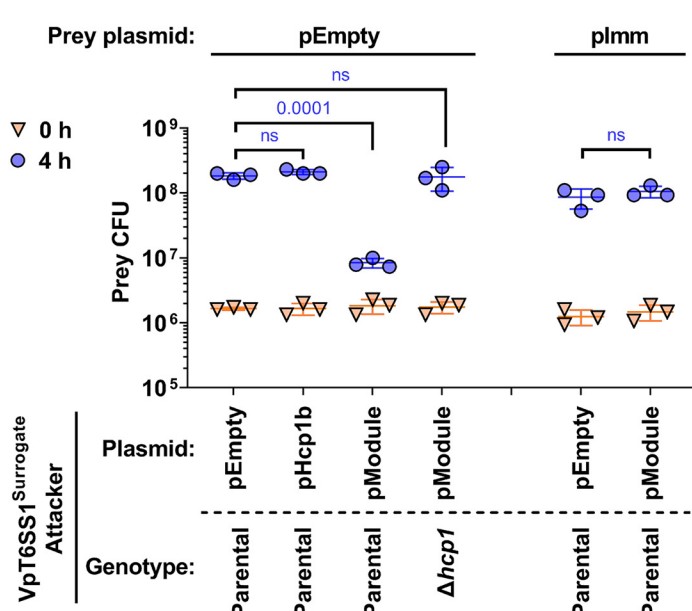

**FIG 3** A DUF4225-containing Hcp auxiliary module contains a T6SS1 effector and immunity pair. (A) Expression (cells) and secretion (media) of VgrG1 from the indicated VpT6SS1[Surrogate] strain or its Δ*hcp1* derivative. Samples were grown in MLB media at 30°C. Loading control (LC), visualized as trihalo compounds' fluorescence of the immunoblot membrane, is shown for total protein lysates. (B) to (C) Viability counts (CFU) of *V. natriegens* prey strain before (0 h) and after (4 h) co-incubation with the surrogate T6SS1 platform strain (VpT6SS1[Surrogate]) or its T6SS1- derivative (Δ*hcp1*) carrying an empty plasmid (pEmpty) or a plasmid for the arabinose-inducible expression of the PoNe DNase-containing VgrG1b module from *V. parahaemolyticus* 12–297/B (pPoNe) (B), of Hcp1b (pHcp1b), or of the three-gene Hcp1b module (pModule) from *V. parahaemolyticus* strain CFSAN018764 (C). In (B), the *V. natriegens* prey strains contain an empty plasmid (pEmpty) or a plasmid for the arabinose-inducible expression of Imm4225[18764] (pImm). The statistical significance between samples at the 4 h time point was calculated using an unpaired, two-tailed Student's *t* test; ns, no significant difference ($P > 0.05$). Data are shown as the mean ± SD; $n = 3$.

DNA from these cells became apparent, manifesting as the fluorescence of propidium iodide (PI), a non-permeable DNA dye that was added to the media, around them (Fig. 4B and Movie S1). These phenotypes are characteristic of cell lysis (57). Sometimes, mostly at later stages of the time course, we observed cells stained from the inside by PI, indicative of the slow permeabilization of the membranes. Similar

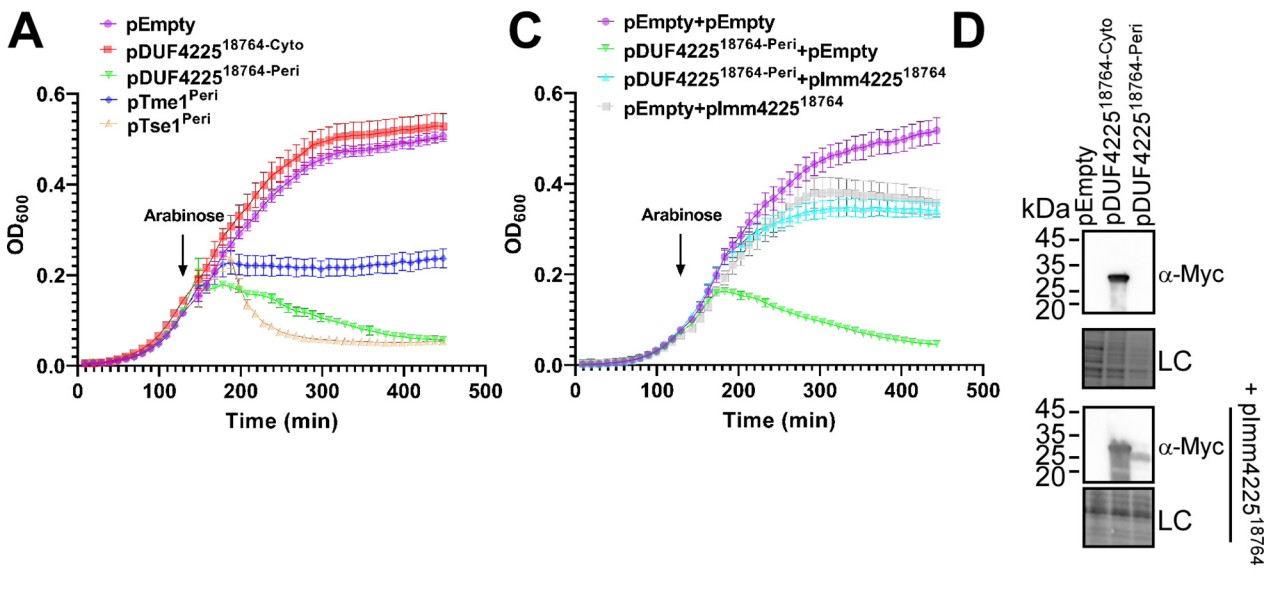

**FIG 4** DUF4225[18764] induces cell lysis upon delivery to the periplasm. (A) and (C) Growth of *E. coli* BL21(DE3) containing arabinose-inducible plasmids, either empty (pEmpty) or expressing the indicated proteins in the cytoplasm (Cyto) or periplasm (Peri; fused to an N-terminal PelB signal peptide). In 'c',

phenotypes were not seen in *E. coli* cells containing an empty expression plasmid (Fig. S4 and Movie S1). Taken together, our results suggest that the activity of DUF4225[18764] in the bacterial periplasm can lead to cell lysis. Importantly, the co-expression of Imm4225[18764] in *E. coli* rescued the cells from the toxicity mediated by periplasmic DUF4225[18764] (Fig. 4C), further supporting its role as the cognate immunity protein of this effector. Notably, as we previously reported for other immunity proteins that are predicted to function in the bacterial membrane or in the periplasm (37), Imm4225[18764] appeared to be mildly toxic when it was over-expressed in *E. coli* (Fig. 4C). The expression of the nontoxic cytoplasmic version of DUF4225[18764] was detected in immunoblots, whereas the toxic periplasmic version of DUF4225[18764] was only detected when Imm4225[18764] was co-expressed (Fig. 4D).

**_Vibrio_ strains without a cognate immunity protein can resist DUF4225[18764] toxicity.** We and others previously reported that certain T6SS effectors appear to have a specific toxicity range; they can intoxicate some but not all bacterial prey strains during T6SS-mediated attacks (54, 58). Since some resistant strains do not carry a homolog of the intoxicating effector's cognate immunity protein, it had been suggested that non-immunity protein-mediated defense mechanisms play a role in such resistance (59). In light of these recent observations, we set out to examine the toxicity range of a T6SS attack mediated by DUF4225[18764]. To this end, we monitored the viability of several marine bacteria prey strains during competition against the VpT6SS1[Surrogate] platform delivering DUF4225[18764]. For prey strains with antibacterial T6SSs known or predicted to be active under the tested conditions, we used mutants in which the T6SS was inactivated by deleting a T6SS core component (e.g., *hcp* or *tssB*), as indicated, to avoid counterattacks during competition. Interestingly, whereas *V. campbellii* ATCC 25920, *V. coralliilyticus* ATCC BAA-450, and *Aeromonas jandaei* DSM 7311 were susceptible to a DUF4225[18764]-mediated attack, *V. parahaemolyticus* 12–297/B was only mildly susceptible to the attack and *V. vulnificus* CMCP6 and *V. parahaemolyticus* RIMD 2210633 were not affected by it (Fig. 5). The 3 latter strains do not carry a homolog of Imm4225[18764]. Importantly, all prey strains except for *V. parahaemolyticus* 12–297/B and *A. jandaei* DSM 7311, which contain a PoNi immunity protein, were susceptible to intoxication by a PoNe DNase effector from *V. parahaemolyticus* strain 12–297/B (7), when it was delivered by the VpT6SS1[Surrogate] platform (Fig. 5); this result indicates that the platform can deliver effectors into the tested prey strains. Therefore, our results reveal that certain bacteria can resist intoxication by DUF4225[18764] even in the absence of a cognate immunity protein.

**DUF4225 is a widespread toxin domain associated with diverse secretion systems.** We next investigated the distribution of the DUF4225 toxin domain in bacterial genomes. We found that the homologs of DUF4225[18764] are widespread in bacterial genomes, almost exclusively belonging to the Pseudomonadota (formerly, Proteobacteria) phylum (Fig. 6A and Data set S4). Interestingly, DUF4225 is found in predicted polymorphic toxins containing N-terminal domains associated with T6SS (e.g., PAAR, VgrG, and Hcp), type V secretion system (e.g., Fil_haemagg_2 and DUF637 [60]), and others (e.g., SpvB [61], RHS_repeat [16], and Sec system signal peptides), although most of the homologs do not contain an identifiable N-terminal domain fused to DUF4225 (Fig. 6B and Data set S4). Sometimes the association of the DUF4225-containing protein with a specific secretion system can be inferred from adjacent genes encoding known components of protein secretion systems, as is the case with DUF4225[18764], which is encoded downstream

**FIG 4** Legend (Continued)
bacteria contain a second plasmid, either empty (pEmpty) or encoding Imm4225[18764]. An arrow denotes the time point at which arabinose (0.05%) was added to the media. (B) Time-lapse microscopy of *E. coli* MG1655 cells stained with Wheat Germ Agglutinin Alexa Fluor 488 conjugate (WGA-488) and that express a periplasmic DUF4225[18764] from an arabinose-inducible plasmid, grown on agarose pads supplemented with chloramphenicol (to maintain the plasmid) and 0.2% arabinose (to induce expression), and propidium iodide (PI). WGA-488 (green), PI (red), phase contrast and merged channels are shown. Size bar = 5 $\mu$m. (D) Expression of the indicated C-terminal Myc-His-tagged DUF4225[18764] variants (Cyto, cytoplasmic; Peri, periplasmic) in *E. coli* BL21(DE3), in the absence (top panels) or presence (bottom panels) of Imm4225[18764] (pImm4225[18764]). DUF4225[18764] proteins were detected by immunoblotting using specific $\alpha$-Myc antibodies. Loading control (LC), visualized as trihalo compounds' fluorescence of the immunoblot membrane, is shown for total protein lysates.

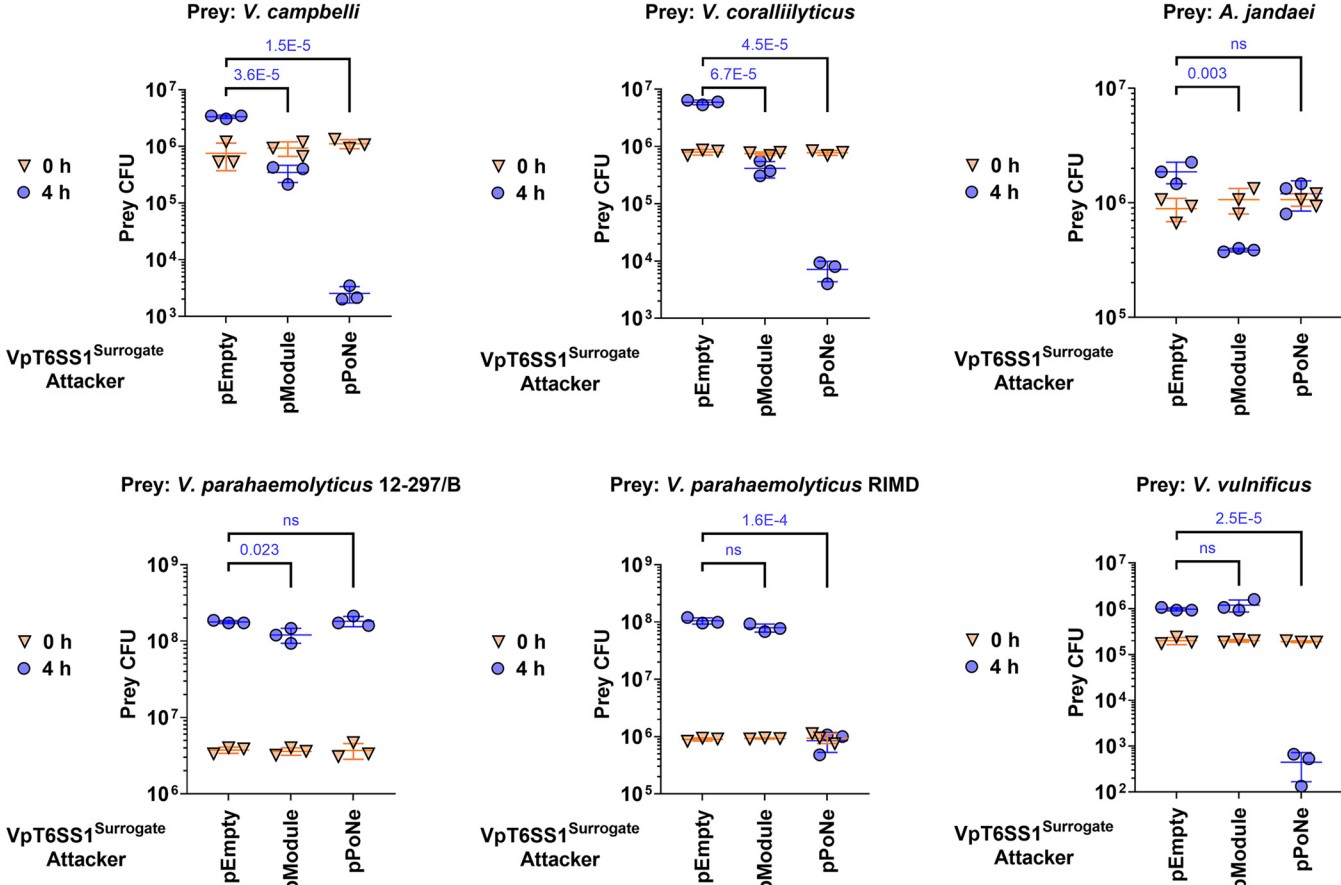

**FIG 5** Varying sensitivity of marine bacteria to DUF4225[18764] T6SS-mediated attacks. Viability counts (CFU) of the indicated prey strains before (0 h) and after (4 h) co-incubation with the surrogate T6SS1 platform strain (VpT6SS1[Surrogate]) carrying an empty plasmid (pEmpty) or a plasmid for the arabinose-inducible expression of the three-gene Hcp1b module (pModule) from *V. parahaemolyticus* strain CFSAN018764 or the PoNe DNase-containing VgrG1b module from *V. parahaemolyticus* 12–297/B (pPoNe). Statistical significance between samples at the 4 h time point was calculated using an unpaired, two-tailed Student's *t* test; ns, no significant difference (*P* > 0.05). Data are shown as the mean ± SD; *n* = 3. The prey strains used were *V. campbellii* ATCC 25920 Δ*hcp1*, *V. coralliilyticus* ATCC BAA-450 Δ*hcp1*, *Aeromonas jandaei* DSM 7311 Δ*tssB*, *V. parahaemolyticus* 12–297/B Δ*hcp1*, *V. parahaemolyticus* RIMD 2210633 Δ*hcp1*, and *V. vulnificus* CMCP6.

of *hcp1b*. Taken together, these results reveal that DUF4225 is a toxin domain that is widespread in secreted polymorphic effectors.

## DISCUSSION

*V. parahaemolyticus* is an emerging marine pathogen responsible for gastroenteritis in humans (40) and for the economically devastating acute hepatopancreatic necrosis disease in shrimp (62). Like many other vibrios, this species employs T6SSs to manipulate and outcompete its rivals (7, 11, 43, 45). Here, we performed a systematic analysis of all available RefSeq *V. parahaemolyticus* genomes, and we revealed the pan-genome repertoire of T6SS gene clusters, putative auxiliary modules, and the predicted effectors therein.

We identified 4 T6SS gene cluster types in *V. parahaemolyticus* genomes. Two systems appear to be ancient and widespread, and 2 seem to have been more recently acquired. T6SS1 and T6SS2, which were previously shown to mediate antibacterial activities (11, 43), are the most common systems in this species. T6SS2 is omnipresent; therefore, it is probably an ancient system that plays an important role in the *V. parahaemolyticus* life cycle. T6SS1, which is present in 68.3% of the genomes, is possibly also an ancient system; however, since T6SS1 appears to be absent from certain lineages (Fig. 2), we propose that it had been lost several times during the evolution of this species. This may be because T6SS1 serves a specialized purpose that is not

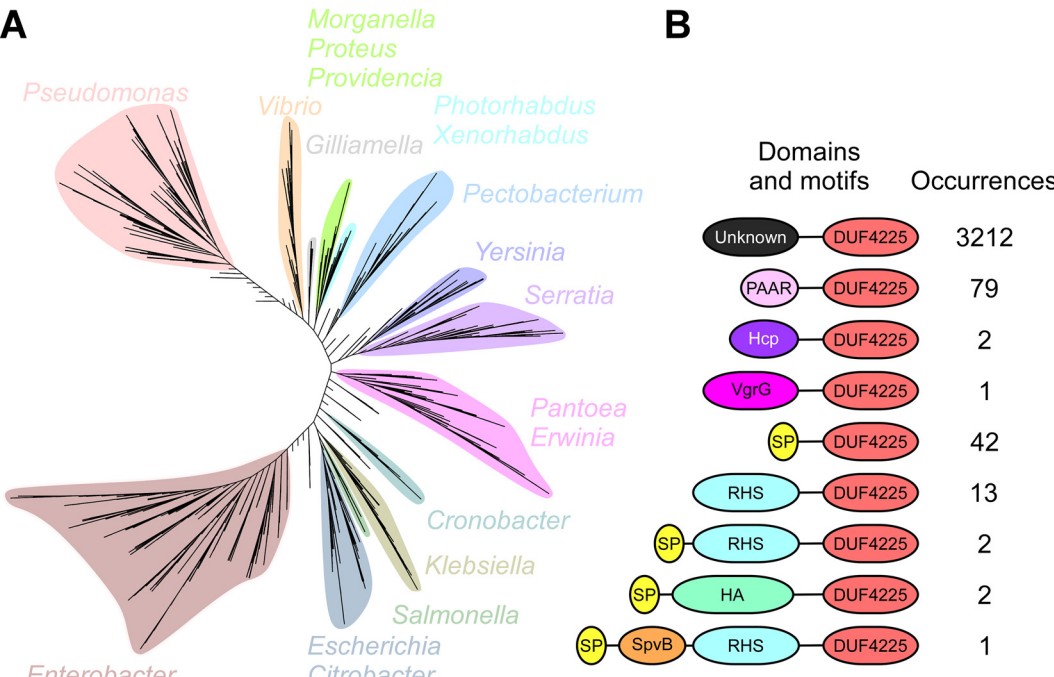

**FIG 6** DUF4225 is a widespread toxin domain. (A) Phylogenetic distribution of bacteria encoding a DUF4225 homolog was based on the DNA sequences of *rpoB* coding for DNA-directed RNA polymerase subunit beta. The evolutionary history was inferred using the neighbor-joining method. (B) The domain architecture and the number of occurrences of DUF4225-containing proteins. SP, signal peptide; HA, Haemagg_act + Fil_haemagg_2 + DUF637. Domain sizes are not to scale.

beneficial for some strains. Indeed, we and others previously proposed that T6SS1 is present predominantly in pathogenic isolates of *V. parahaemolyticus* (41–43); therefore, it is plausible that nonpathogenic strains have lost this system, since it may only be required during colonization of a host. T6SS3 and T6SS4 are found in <3% of the genomes combined. Since DNA mobility genes flank these gene clusters, we propose that they have been recently acquired via HGT. This seems possible considering the ability of vibrios to horizontally acquire large DNA fragments (46, 63, 64). The role and activity mediated by these 2 systems remains to be investigated. Nevertheless, we propose that T6SS3 mediates anti-eukaryotic activity, since it is similar to the previously reported T6SS3 in *V. proteolyticus* (50, 51), and since we identified a predicted effector within the cluster that lacks an identifiable potential immunity protein, suggesting that it does not mediate antibacterial activity. We also propose that T6SS4 mediates antibacterial activities, since we identified predicted antibacterial effectors within it, which have a downstream adjacent gene that possibly encodes a cognate immunity protein (Fig. 1B and Table 1).

Our analysis also revealed diverse putative T6SS auxiliary modules encoding at least one of the hallmark secreted proteins, Hcp, VgrG, and PAAR. In most of these putative auxiliary modules, we found known or predicted effectors that can diversify the toxic repertoires of this species' T6SSs. The identity of these genes as effectors is supported by their homology to known effectors, by their position downstream of genes encoding T6SS adaptors, by the presence of domains associated with T6SS-secreted proteins (e.g., MIX (36) and FIX (7)), and by the presence of homologs of their C-terminus in polymorphic toxins (Table 1). Interestingly, several predicted effectors, identified within auxiliary modules, do not resemble previously studied toxins, and they may therefore employ novel mechanisms of action. Notably, although this was not directly addressed in this work, additional orphan effectors that are not encoded within T6SS gene clusters or within auxiliary modules next to a hallmark secreted T6SS component have been previously reported in *V. parahaemolyticus* genomes (7, 11, 42, 45); many are

found next to DNA mobility genes, suggesting that they may also be horizontally shared within this species (42, 48). These orphan effectors further diversify the T6SS effector repertoire of this species.

In this study, we investigated the role of a gene encoding a DUF4225-containing protein; we showed that it is an antibacterial T6SS1 effector, and that its cognate immunity protein is encoded directly downstream. To the best of our knowledge, this is the first activity described for this domain of unknown function. Notably, DUF4225-encoding genes are common in Hcp-containing auxiliary modules, suggesting that they too are T6SS effectors. Our analysis also revealed that DUF4225 is widespread outside vibrios, where it is associated with T6SSs, as well as with other secretion systems that deliver polymorphic toxins.

We found that DUF4225 exerts its toxic activity in the bacterial periplasm. However, its mechanism of action and its cellular target remain unknown. Since its expression in the periplasm of *E. coli* cells led to cell lysis, we hypothesize that it affects the stability of the peptidoglycan or the membrane. Future biochemical and structural work will be required to address these open questions. Interestingly, we identified marine bacteria that resist intoxication by a DUF4225-containing effector. Since we did not identify homologs of the cognate immunity protein encoded within the genomes of these resistant strains, we predict that they employ a yet-unknown non-immunity protein-mediated defense mechanism that counteracts the toxicity of this effector.

The effectorless surrogate T6SS platform constructed in this work is an important tool, allowing us to rapidly identify and investigate T6SS effectors encoded by any *V. parahaemolyticus* strain. Although we previously reported the construction of a similar surrogate system (11), the effectorless version reported here is superior: (i) the toxicity mediated by a predicted effector can be tested against diverse prey strains, thus reducing the possibility of false negatives due to the lack of toxicity against a *V. parahaemolyticus* RIMD 2210633 prey (in hindsight, DUF4225[18764] would have been a false negative in the previous version of the surrogate platform); and (ii) no endogenous cargo effectors compete with the investigated predicted effector for loading onto the secreted tube and spike, thus increasing the probability that it will be delivered by the surrogate system.

In conclusion, we present the first comprehensive analysis of the T6SS repertoire in the *V. parahaemolyticus* pan-genome. Our results reveal 4 T6SSs found within this species; they also indicate that mobile auxiliary modules probably contribute greatly to diversifying the T6SS effector repertoires in various strains. We also describe a role for the widespread DUF4225 as an antibacterial toxin domain, and we identify additional putative effectors that await investigation.

## MATERIALS AND METHODS

**Strains and Media.** For a complete list of strains used in this study, see Table S1. *E. coli* strains were grown in 2xYT broth (1.6% wt/vol tryptone, 1% wt/vol yeast extract, and 0.5% wt/vol NaCl) or on Lysogeny Broth (LB) agar plates containing 1% wt/vol NaCl at 37°C, or at 30°C when harboring effector expression plasmids. The media were supplemented with chloramphenicol (10 $\mu$g/mL) or kanamycin (30 $\mu$g/mL) to maintain plasmids, and with 0.4% wt/vol glucose to repress protein expression from the arabinose-inducible promoter, P*bad*. To induce expression from P*bad*, L-arabinose was added to the media at 0.1–0.2% (wt/vol), as indicated.

*Vibrio parahaemolyticus*, *V. natriegens*, *V. coralliilyticus*, and *V. vulnificus* strains were grown in MLB media (LB media containing 3% wt/vol NaCl) or on marine minimal media (MMM) agar plates (1.5% wt/vol agar, 2% wt/vol NaCl, 0.4% wt/vol galactose, 5 mM MgSO$_4$, 7 mM K$_2$SO$_4$, 77 mM K$_2$HPO$_4$, 35 mM KH$_2$PO$_4$, and 2 mM NH$_4$Cl). *V. campbellii* were grown in MLB media and on MLB agar plates. *A. jandaei* were grown in LB media and on LB agar plates. When vibrios or *A. jandaei* contained a plasmid, the media were supplemented with kanamycin (250 $\mu$g/mL), chloramphenicol (10 $\mu$g/mL), or streptomycin (100 $\mu$g/mL) to maintain the plasmid. Bacteria were grown at 30°C. To induce expression from P*bad*, L-arabinose was added to media at 0.05% (wt/vol).

**Plasmid construction.** For a complete list of plasmids used in this study, see Table S1. The DNA sequence of the Hcp1b auxiliary module from *V. parahaemolyticus* strain CFSAN018764 (positions 296 to 27 in NZ_LHBG01000025.1) was synthesized by Twist Bioscience (USA). The entire module sequence or the sequences of genes within it were PCR amplified and then inserted into the multiple cloning site (MCS) of pBAD^K/Myc-His, pPER5, or pBAD33.1^F, in-frame with C-terminal Myc-His or FLAG tag, using the Gibson assembly method (65). For the expression of Imm4225[18764] in *V. natriegens*, the region spanning

the *araC* gene to the rrnT1 terminator was amplified from pBAD33.1[F] containing the gene in its MCS, and then inserted into the NotI restriction site of pCLTR plasmid using restriction-digestion and ligation.

Plasmids were introduced into *E. coli* using electroporation. Transformants were grown on agar plates supplemented with 0.4% wt/vol glucose to repress expression from the P*bad* promoter during the subcloning steps. Plasmids were introduced into vibrios and *A. jandaei* via conjugation. Trans-conjugants were grown on MLB agar plates for *V. campbellii*, LB agar plates for *A. jandaei*, or MMM agar plates for all other vibrios. Plates were supplemented with appropriate antibiotics to maintain the plasmids.

**Construction of deletion strains.** For in-frame deletions of *hcp1* in *V. coralliilyticus* (VIC_RS16330) or of *hcp1* in *V. campbellii* (A8140_RS17660), 1 kb sequences upstream and downstream of each gene were subcloned into pDM4, a Cm[r]OriR6K suicide plasmid (66). Next, pDM4 constructs were introduced into the respective *Vibrio* strain via conjugation. Trans-conjugants were selected on MMM agar plates containing chloramphenicol (10 $\mu$g/mL). The resulting trans-conjugants were grown on MMM agar plates containing sucrose (15% wt/vol) for counter-selection and loss of the SacB-containing pDM4.

The VpT6SS1[Surrogate] strain and its $\Delta hcp1$ derivative were generated by consecutive deletions or mutations of the relevant genes using previously reported pDM4 plasmids.

**Toxicity assays in *E. coli*.** To assess the toxicity mediated by DUF4225[18764], pBAD[K]/Myc-His (for cytoplasmic expression) and pPER5 (for periplasmic expression fused to an N-terminal PelB signal peptide) plasmids, either empty or encoding DUF4225[18764] were transformed into *E. coli* BL21(DE3). *E. coli* transformants were grown overnight in 2xYT media supplemented with kanamycin (30 $\mu$g/mL) under P*bad* repressing conditions (0.4% wt/vol glucose). Overnight cultures were washed to remove residual glucose, and normalized to $OD_{600} = 0.01$ in 2xYT media supplemented with kanamycin. Then, 200 $\mu$L of each bacterial culture were transferred into 96-well plates in quadruplicate. The cultures were grown at 37°C with agitation (205 cpm) in a microplate reader (BioTek SYNERGY H1). After 2 h of growth, L-arabinose was added to each well at a final concentration of 0.1% (wt/vol), to induce protein expression. $OD_{600}$ readings were recorded every 10 min for 7 h.

To test the ability of Imm4225[18764] to antagonize the toxicity of DUF4225[18764], a pBAD33.1[F] plasmid, either empty or encoding Imm4225[18764], was co-transformed with a pPER5 plasmid, either empty or encoding DUF4225[18764], into *E. coli* BL21(DE3). The growth of these strains was determined as described above. Growth assays were performed at least four times with similar results. Results from a representative experiment are shown.

**Protein expression in *E. coli*.** To determine the expression of C-terminal Myc-His-tagged DUF4225[18764], *E. coli* BL21(DE3) bacteria carrying a single arabinose-inducible expression plasmid, either empty or encoding a cytoplasmic or a periplasmic DUF4225[18764], or bacteria carrying two plasmids, one for expression of Imm4225[18764] and the other either empty or expressing the periplasmic version of DUF4225[18764], were grown overnight in 2xYT media supplemented with the appropriate antibiotics to maintain plasmids, and glucose to repress expression from P*bad*. The cultures were washed twice with fresh 2xYT medium to remove residual glucose, and then diluted 100-fold in 5 mL of fresh 2xYT medium supplemented with appropriate antibiotics and grown for 2 h at 37°C. To induce protein expression, 0.1% (wt/vol) L-arabinose was added to the media. After 4 additional hours of growth at 30°C, 1.0 $OD_{600}$ units of cells were pelleted and resuspended in 100 $\mu$L of (2X) Tris-Glycine SDS sample buffer (Novex, Life Sciences). Samples were boiled for 5 min, and cell lysates were resolved on Mini-PROTEAN TGX Stain-Free precast gels (Bio-Rad). For immunoblotting, $\alpha$-Myc antibodies (Santa Cruz Biotechnology, 9E10, mouse MAb; sc-40) were used at 1:1,000 dilution. Protein signals were visualized in a Fusion FX6 imaging system (Vilber Lourmat) using enhanced chemiluminescence (ECL) reagents. Experiments were performed at least three times with similar results; the results from representative experiments are shown.

**Bacterial competition assays.** Bacterial competition assays were performed as previously described (37), with minor changes. Briefly, cultures of the indicated attacker and prey strains were grown overnight. Bacterial cultures were then normalized to $OD_{600} = 0.5$ and mixed at a 10:1 (attacker: prey) ratio. The mixtures were spotted (25 $\mu$L) on MLB agar plates supplemented with 0.05% (wt/vol) L-arabinose, and incubated for 4 h at 30°C. CFU/of the prey strains were determined at the 0 and 4-h time points. The experiments were performed at least three times with similar results. Results from a representative experiment are shown.

**Fluorescence microscopy.** Cell morphology and membrane permeability during the expression of DUF4225[18764] in *E. coli* was assessed as previously described (37). Briefly, overnight-grown *E. coli* MG1655 cells carrying a pPER5 plasmid, either empty or encoding DUF4225[18764], were diluted 100-fold into 5 mL of fresh LB media supplemented with kanamycin and 0.2% (wt/vol) glucose. Bacterial cultures were grown for 2 h at 37°C, and then cells were washed with 0.15 M NaCl to remove residual glucose. Bacterial cultures were normalized to $OD_{600} = 0.5$ in 0.15 M NaCl solution. To visualize the cell wall of *E. coli*, 20 $\mu$L of bacterial cultures were incubated with Wheat Germ Agglutinin Alexa Fluor 488 Conjugate (Biotium; Catalogue no. 29022–1) at a final concentration of 0.1 mg/mL, and incubated for 10 min at room temperature (RT). Next, 1 $\mu$L was spotted on LB agarose pads (1% wt/vol agarose supplemented with 0.2% wt/vol L-arabinose) onto which 1 $\mu$L of the membrane-impermeable DNA dye, propidium iodide (PI; 1 mg/mL; Sigma-Aldrich) had been pre-applied. After the spots had dried (1–2 min at RT), the agarose pads were mounted, facing down, on 35 mm glass bottom CELLview cell culture dishes (Greiner). Cells were then imaged every 5 min for 4 h under a fluorescence microscope, as detailed below. The stage chamber (Okolab) temperature was set to 37°C. Bacteria were imaged in a Nikon Eclipse Ti2-E inverted motorized microscope equipped with a CFI PLAN apochromat DM 100X oil lambda PH-3 (NA, 1.45) objective lens, a Lumencor SOLA SE II 395 light source, and ET-EGFP (#49002, used to visualize the Alexa Fluor 488 signal), and an RFP filter cube (#49005, used to visualize

the PI signal) filter sets. Images were acquired using a DS-QI2 Mono cooled digital microscope camera (16 MP) and were post-processed using Fiji ImageJ suite. The experiments were performed three times. Results from a representative experiment are shown.

**VgrG1 secretion assays.** *V. parahaemolyticus* VpT6SS1<sup>Surrogate</sup> and its Δ*hcp1* derivative strain were grown overnight at 30°C in MLB media. Bacterial cultures were normalized to $OD_{600}$ = 0.18 in 5 mL of MLB media, and after 5 h of incubation at 30°C with agitation (220 rpm), expression fractions (cells) and secretion fractions (media) were collected and processed as previously described (37).

**Identifying T6SS gene clusters in *V. parahaemolyticus*.** A local database containing the RefSeq bacterial nucleotide and protein sequences was generated (last updated on June 11, 2022). *V. parahaemolyticus* genomes were retrieved from the local database and OrthoANI was performed as described previously (11, 67). Two genomes (assembly accessions GCF_000591535.1 and GCF_003337295.1) with OrthoANI values <95% were removed from the data set.

The presence of T6SS gene clusters in *V. parahaemolyticus* genomes was determined by following the two-step procedure described below. In the first step, BLASTN (68) was employed to align *V. parahaemolyticus* nucleotide sequences against the nucleotide sequences of representative T6SS clusters (Fig. 1 and Data set S2). The expect value threshold was set to $10^{-12}$ and the minimal alignment length was 500 bp. The results were then sorted by their nucleotide accession numbers and bit score values (from largest to smallest), and the best alignments for each nucleotide accession number were saved. This step resulted in a list of *V. parahaemolyticus* nucleotide accession numbers and their best alignments to the representative T6SS gene clusters, including the positions of the alignments. In the second step, a two-dimensional matrix was generated for each T6SS gene cluster in which rows represented the *V. parahaemolyticus* genomes and columns represented the coordinates of the specific T6SS gene cluster. The matrices were then filled in with the percent identity values, based on the positions of the alignments. Finally, the overall coverage was calculated for each T6SS gene cluster in each genome. *V. parahaemolyticus* genomes with at least 70% overall coverage of T6SS gene cluster were regarded as containing that T6SS gene cluster (Data set S2).

**Identifying T6SS auxiliary modules.** RPS-BLAST (69) was employed to identify proteins containing Hcp (COG3157), PAAR (DUF4150, PAAR_motif, PAAR_1, PAAR_2, PAAR_3, PAAR_4, PAAR_5, PAAR_RHS, PAAR_CT_1, PAAR_CT_2), and VgrG (COG3501) domains that were retrieved from the Conserved Domain Database (70), in *V. parahaemolyticus* genomes. Protein accessions located at the ends of contigs were removed. T6SS auxiliary modules were manually identified, based on the distance from the T6SS gene clusters, the genomic architecture, and the conserved domains in neighboring genes (Data set S3).

**Identifying DUF4225 homologs with domain and neighborhood analysis.** The Position-Specific Scoring Matrix (PSSM) of the DUF4225 domain was reconstructed using amino acids 105–243 of DUF4225<sup>18764</sup> from *V. parahaemolyticus* strain CFSAN018764 (WP_065788326.1). Five iterations of PSI-BLAST were performed against the reference protein database (a maximum of 500 hits with an expect value threshold of $10^{-6}$ and a query coverage of 70% were used in each iteration). RPS-BLAST was then performed to identify DUF4225-containing proteins. The results were filtered using an expect value threshold of $10^{-8}$ and a minimal coverage of 70%. The genomic neighborhoods of DUF4225-containing proteins were analyzed as described previously (11, 45). Duplicated protein accessions appearing in the same genome in more than one genomic accession were removed if the same downstream protein existed at the same distance (Data set S4).

**Constructing phylogenetic trees.** The nucleotide sequences of *rpoB* coding for DNA-directed RNA polymerase subunit beta were retrieved from the local RefSeq database. Partial and pseudogene sequences were not included in the analyses. In the case of bacterial genomes encoding DUF4225 homologs, the *rpoB* sequences were first clustered using CD-HIT to remove identical sequences (100% identity threshold). Phylogenetic analyses of bacterial genomes were conducted using the MAFFT 7 server (mafft.cbrc.jp/alignment/server/). The *rpoB* sequences were aligned using MAFFT v7 FFT-NS-2 (71, 72). In the case of *V. parahaemolyticus* genomes, the evolutionary history was inferred using the neighbor-joining method (73) with the Jukes-Cantor substitution model (JC69). The analysis included 1,694 nucleotide sequences and 3,964 conserved sites. In the case of bacterial genomes encoding DUF4225, the evolutionary history was inferred using the average linkage (UPGMA) method and included 2,816 nucleotide sequences.

The protein accessions of TssB and Hcp from *V. parahaemolyticus* genomes were retrieved and unique sequences were aligned using CLUSTAL Omega (74). The evolutionary history was inferred using the Neighbor-Joining method (73). The analysis of TssB included 42 amino acid sequences and 166 conserved sites. The analysis of Hcp included 27 amino acid sequences and 133 conserved sites. Evolutionary analyses in both cases were conducted in MEGA X (75).

**Identifying effectors in T6SS gene clusters and auxiliary modules.** The presence of effectors in T6SS gene clusters and auxiliary modules was determined by homology to previously studied effectors in *V. parahaemolyticus*, by the location within auxiliary modules downstream of secreted core components (i.e., Hcp, VgrG, or PAAR) or of known T6SS adaptor-encoding genes (i.e., DUF4123, DUF1795, or DUF2169), and by the presence of potential C-terminal toxin domains identified using NCBI's Conserved Domain Database (76). The presence of homologs of C-terminal toxin domains in other polymorphic toxin classes was determined using Jackhmmer (77).

## SUPPLEMENTAL MATERIAL

Supplemental material is available online only.

**DATA SET S1**, XLSX file, 0.3 MB.

**DATA SET S2**, XLSX file, 8 MB.
**DATA SET S3**, XLSX file, 0.3 MB.
**DATA SET S4**, XLSX file, 5.4 MB.
**MOVIE S1**, AVI file, 4.6 MB.
**FIG S1**, TIF file, 0.8 MB.
**FIG S2**, TIF file, 0.5 MB.
**FIG S3**, TIF file, 0.8 MB.
**FIG S4**, TIF file, 2.8 MB.
**TABLE S1**, DOCX file, 0.03 MB.

## ACKNOWLEDGMENTS

This project received funding from the European Research Council under the European Union's Horizon 2020 research and innovation program (grant agreement no. 714224 to D. Salomon), and from the Israel Science Foundation (grant no. 920/17 to D. Salomon, and grant no. 1362/21 to D. Salomon and E. Bosis). C.M. Fridman was supported by scholarships from the Clore Israel Foundation and from the Manna Center Program in Food Safety and Security at Tel Aviv University, as well as by a scholarship for outstanding doctoral students from the Orthodox community from the Council for Higher Education.

We thank Hila Saar, Nikol Dvir, and Nely Altshul for their excellent technical assistance, and members of the Salomon and Bosis laboratories for helpful discussions and suggestions.

B.J., E.B., and D.S. conceptualized, investigated, and ensured methodology. B.J. wrote, reviewed, and edited the manuscript. K.K. and C.M.F. investigated and ensrued methodology. E.B. and D.S. acquired funds and wrote the original draft. D.S. supervised the study.

We declare that there are no conflicts of interest.

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
