## [Reviewer comments · mSystems]

Multiple T6SSs, mobile auxiliary modules, and effectors revealed in a systematic analysis of the *Vibrio parahaemolyticus* pan-genome

Biswanath Jana, Kinga Keppel, Chaya Fridman, Eran Bosis, and Dor Salomon

Corresponding Author(s): Dor Salomon, Tel Aviv University

Review Timeline:

Submission Date:	August 2, 2022
Editorial Decision:	September 12, 2022
Revision Received:	September 21, 2022
Accepted:	September 22, 2022

Editor: Seth Bordenstein

Reviewer(s): Disclosure of reviewer identity is with reference to reviewer comments included in decision letter(s). The following individuals involved in review of your submission have agreed to reveal their identity: Thandavarayan Ramamurthy (Reviewer #3)

Transaction Report:

DOI: <https://doi.org/10.1128/msystems.00723-22>

September 12, 2022

Dr. Dor Salomon
Tel Aviv University
Clinical Microbiology and Immunology
Tel Aviv
Israel

Re: mSystems00723-22 (Multiple T6SSs, mobile auxiliary modules, and effectors revealed in a systematic analysis of the *Vibrio parahaemolyticus* pan-genome)

Dear Dr. Dor Salomon:

Thank you for submitting your manuscript to mSystems. We have completed our review with two reviewers and I am pleased to inform you and your team that, in principle, we expect to accept it for publication in mSystems. However, acceptance will not be final until you have adequately addressed the reviewer comments below.

Preparing Revision Guidelines

Sincerely,

Seth Bordenstein

Editor, mSystems

Journals Department
Reviewer comments:

Reviewer #1 (Comments for the Author):

Vibrio parahaemolyticus is an important bacterial pathogen of both humans and marine animals that uses a type VI secretion system to deliver cocktails of toxin protein effectors into the cytoplasm or periplasm of neighboring cells. Many bacteria use a T6SS to compete with rival cells. The authors deploy a method to analyse >1700 *Vibrio parahaemolyticus* genomes. They identify a large number of putative effectors not described prior, and then demonstrated experimentally that DUF4225 is indeed a T6 effector and Imm4225 its immunity factor. They then identify many other proteins carry a DUF4225 domain, suggesting that this toxin is widespread. The paper is well written and experiments conducted properly. It is impactful as it expands the repertoire of T6SS effectors described in this bacterium. My major comment regards the lack of distinction in the text between effectors that are predicted to have activity, versus though like DUF4225 shown her experimentally to have activity.

Presumably lines 57-70 in the Introduction are referring to T6SSs in general, but this should be made clear. Have studies of *Vibrio parahaemolyticus* pathogenic isolates demonstrated a contribution of the T6SS1 to colonization or disease? This would be useful for supporting the importance of this system to *Vibrio parahaemolyticus* biology.

Lines 110-11: How do you determine the boundary of the T6SS2 cluster that allows you to conclude there are no known or predicted effectors within it? Just to get the accounting correct - there are zero T6SS1 effectors and two for T6SS2 based on work here and by others prior? The impact of identifying auxiliary modules and their effectors later would be strengthened if it was cleared stated here how few effectors have been found to date either within or outside of T6SS1 and T6SS2.

Throughout the text, it is often stated that something is an auxiliary module or is an effector based solely on homology or annotation. Without demonstration of activity, I do not think this is appropriate. In many instances "putative" or "predicted" is used. For example, lines 151-2 "putative cargo effectors" and "Table 1 Predicted effectors". But this is inconsistent throughout.

Line 134: "We found diverse auxiliary module types" How is it known that these are in fact auxiliary modules that encode one or more effectors that are used by one or more of the T6SSs in that isolate? Is it possible that there are genes similar enough in sequence to an *hcp*, *vgrG* or PAAR to be identified by your method but that are not in fact T6SS genes? Is it possible that some of the Auxiliary modules indicated in Figure 2 are not T6SS genes? This distinction between the large set of predicted auxiliary effectors and those experimentally validated is critical. This manuscript proceeds to validate that a DUF4225 predicted to encode an effector an immunity protein indeed do so.

Line 138: Similarly, the statement "secreted T6SS component" requires demonstration that the gene product is secreted in a T6SS-dependent manner.

Fig 3. It is interesting that the chromosomal *Hcp1*, but not the *Hcp* in the Module, appears to be required for killing with the effector encoded in the Module. I did not see mention of the results in Figure 3 describing the Prey CFU when the Attacker genotype is null for *hcp1*. This seems particularly important, especially since the presence of an *hcp* was used to identify this and other putative auxiliary modules. If the *hcp* is dispensable, might that suggest there can be orphan effector immunity pairs to be found that are not adjacent to an *hcp*, (or *vgrG* or PAAR)?

Fig 4. In panel C, it is notable that the expression of Imm4225 with the pEmpty impairs growth. Can a comment be made regarding this result?

Lines 337-40. To construct the "effectorless surrogate T6SS platform" two effectors were deleted, but the Vp1415 is a two aa substitution (lines 177-8), which lacks activity. I am confused, because wouldn't the Vp1415 still be loaded and delivered but not have activity in a neighbor? Has this been shown here or prior? Wouldn't a strain expressing a heterologous effector still experience competition for loading? Clarification here would be useful.

Reviewer #3 (Comments for the Author):

The type VI secretion system (T6SS) is mostly present in Gram-negative bacteria and has been used to compete with the host and helps in the invasion process. The *Hcp* protein bordered with the *VgrG*/PAAR complex is vital in antibacterial or anti-eukaryotic effectors transportation. The genomic backbone and the functional aspects of T6SS might vary among several bacterial species. In this study, the authors focused on the pan-genome makeup of *Vibrio parahaemolyticus* to better understand the ability of T6SS potential. This study was planned and executed well to address several research questions.

Following are the comments that I would like to set forth.

1. Several abbreviations need to be spelled out in its first use (e.g. PAAR, H-NS, etc.).
2. Line 120. Mention here the specific virulence function of anti-eukaryotic activity in relation to disease progression.
3. Line 133. The VgrG effector and its alleles are widely reported in potential pathogens like *V. cholerae*, *P. aeruginosa* etc. A comparative account of this effector would be an interesting information.
4. *V. parahaemolyticus* has a T6SS both on chromosome-1 (T6SS1) and chromosome 2 (T6SS2). Interestingly, the authors have identified two additional T6SSs namely T6SS3 and T6SS4. However, the advantage of having these two new T6SSs to the bacterium has not been discussed in evolutionary or virulence prospective.
5. While discussion cognate immunity protein, the authors should consider the other potential pathogenic vibrios like *V. cholerae*, which occupies the same milieu, while causing the disease.
6. It has been mentioned that two additional T6SSs has been recently acquired by *V. parahaemolyticus*. As the authors well aware that pandemic strains of this pathogen (serovars O3:K6, O4:K68, etc.) have emerged and spread across the globe causing several outbreaks. It would be interesting to investigate and discuss about these new T6SSs in the pandemic strains. I suggest to screen the pandemic strain WGSs to check the preferential selection of T6SS3 and T6SS4 if any.

Point-by-point reply to reviewer comments

Reviewer #1 (Comments for the Author):

Vibrio parahaemolyticus is an important bacterial pathogen of both humans and marine animals that uses a type VI secretion system to deliver cocktails of toxin protein effectors into the cytoplasm or periplasm of neighboring cells. Many bacteria use a T6SS to compete with rival cells. The authors deploy a method to analyse >1700 *Vibrio parahaemolyticus* genomes. They identify a large number of putative effectors not described prior, and then demonstrated experimentally that DUF4225 is indeed a T6 effector and Imm4225 its immunity factor. They then identify many other proteins carry a DUF4225 domain, suggesting that this toxin is widespread. The paper is well written and experiments conducted properly. It is impactful as it expands the repertoire of T6SS effectors described in this bacterium. My major comment regards the lack of distinction in the text between effectors that are predicted to have activity, versus though like DUF4225 shown her experimentally to have activity.

Presumably lines 57-70 in the Introduction are referring to T6SSs in general, but this should be made clear.

To clarify that this section refers to T6SSs in general, we rephrased the first sentence. It now reads: “Many T6SS effector families that contain toxin domains mediating antibacterial activities have been reported in various bacteria.”

Have studies of *Vibrio parahaemolyticus* pathogenic isolates demonstrated a contribution of the T6SS1 to colonization or disease? This would be useful for supporting the importance of this system to *Vibrio parahaemolyticus* biology.

We are not aware of studies that directly assess the contribution of T6SS1 to *V. parahaemolyticus* colonization or disease during infection of a host. Nevertheless, a previous comparative transcriptomics (RNAseq) study indicated that T6SS1 is induced during infection of rabbit intestine (Livny et al., *Nucleic Acids Res.* 42, (2014)) suggesting that T6SS1 may play a role during infection. Notably, the importance of T6SS1 to *V. parahaemolyticus* biology is emphasized by our current findings showing that it is widely distributed and well conserved within this species.

Lines 110-11: How do you determine the boundary of the T6SS2 cluster that allows you to conclude there are no known or predicted effectors within it?

The reviewer raises an important question. The boundaries of T6SS2 were previously predicted by others (e.g., Zhang et al, *Archives of Microbiology*, 2017; <https://doi.org/10.1007/s00203-017-1361-6>). We confirmed these boundaries by

identifying the T6SS2 gene cluster's flanking genes in other vibrios, where they flank a different set of gene (e.g., in *Vibrio campbellii* ATCC BAA-1116 chromosome II, GenBank: CP006606.1; Locus tags M892_23370 and M892_23605 correspond to VP_RS20080 and VP_RS20210 in *V. parahaemolyticus* RIMD 2210633). Although VP_RS20200 and VP_RS20205 are not found in this location in *V. campbellii*, they appear to be part of an operon together with the T6SS2 flanking gene, and are therefore not predicted to be part of the T6SS2 gene cluster. In the revised manuscript, we added VP_RS20190 and VP_RS20195 to the schematic representation of T6SS2 in Fig. 1b. These two genes are predicted by HHpred analysis to be response regulators.

All of the genes within the T6SS2 of the reference strain RIMD 2210633 account for known or predicted core and accessory T6SS components, or regulatory components (as denoted in Fig. 1b), leading us to conclude that none of these genes is a T6SS effector. Nevertheless, although manual inspection of a few dozen T6SS2 gene clusters revealed very conserved gene sequences and gene composition, we cannot rule out the possibility of derivative T6SS2 gene clusters that may contain additional genes that encode effectors. Therefore, we revised the relevant text, which now reads: "We did not identify known or potential effectors encoded within the T6SS2 gene cluster of the *V. parahaemolyticus* reference strain RIMD 2210633, nor in T6SS2 gene clusters of several other strains, which were manually assessed".

Just to get the accounting correct - there are zero T6SS1 effectors and two for T6SS2 based on work here and by others prior? The impact of identifying auxiliary modules and their effectors later would be strengthened if it was clearly stated here how few effectors have been found to date either within or outside of T6SS1 and T6SS2.

Regarding prior knowledge of *V. parahaemolyticus* T6SS effectors:

T6SS2 - No effectors have been associated with T6SS2 to date, although we previously described effectors of a similar system in *V. alginolyticus* strain 12G01 (Salomon et al., PLoS Pathogens, 2014). This is noted in the text – "T6SS2 was recently shown to mediate antibacterial activities; however, its effector repertoire remains unknown."

T6SS1 – two conserved effectors are encoded within the T6SS1 gene cluster, one at each end, while different isolates carry diverse repertoires of accessory T6SS1 effectors. This had been clarified in the text: "T6SS1 deploys two conserved antibacterial E/I pairs, which are encoded by the main T6SS1 gene cluster, as well as "accessory" E/I pairs that differ between isolates and diversify the effector repertoire (36, 45). To date, few accessory T6SS1 E/I pairs were found in auxiliary T6SS modules containing a gene encoding VgrG (7), or as orphan operons that often reside next to DNA mobility elements (11, 36, 42). Only a handful of these accessory T6SS1 E/I pairs have been experimentally validated".

Throughout the text, it is often stated that something is an auxiliary module or is an

effector based solely on homology or annotation. Without demonstration of activity, I do not think this is appropriate.

To convey the caveat that the reviewer raises to the readers, we changed the text in several places, both in the Results section and the Discussion section, to clarify that we identified putative modules. For example: “Therefore, to identify putative auxiliary T6SS modules, we searched the *V. parahaemolyticus* pan-genome for Hcp, VgrG, and PAAR... We found diverse putative auxiliary module types...”; and “Our analysis also revealed diverse putative T6SS auxiliary modules encoding at least one of the secreted proteins, Hcp, VgrG, and PAAR”.

In many instances "putative" or "predicted" is used. For example, lines 151-2 "putative cargo effectors" and "Table 1 Predicted effectors". But this is inconsistent throughout.

We changed all the instances relating to effectors to read “predicted effectors”.

Line 134: "We found diverse auxiliary module types" How is it known that these are in fact auxiliary modules that encode one or more effectors that are used by one or more of the T6SSs in that isolate? Is it possible that there are genes similar enough in sequence to an hcp, vgrG or PAAR to be identified by your method but that are not in fact T6SS genes? Is it possible that some of the Auxiliary modules indicated in Figure 2 are not T6SS genes? This distinction between the large set of predicted auxiliary effectors and those experimentally validated is critical. This manuscript proceeds to validate that a DUF4225 predicted to encode an effector an immunity protein indeed do so.

As mentioned above, we now clarify throughout the text that the modules we describe are putative, and that the effectors therein are predicted.

Line 138: Similarly, the statement "secreted T6SS component" requires demonstration that the gene product is secreted in a T6SS-dependent manner.

We changed the text to read: “Notably, some modules contain more than one predicted hallmark secreted T6SS component...”

Fig 3. It is interesting that the chromosomal Hcp1, but not the Hcp in the Module, appears to be required for killing with the effector encoded in the Module. I did not see mention of the results in Figure 3 describing the Prey CFU when the Attacker genotype is null for hcp1. This seems particularly important, especially since the presence of an hcp was used to identify this and other putative auxiliary modules. If the hcp is dispensable, might that suggest there can be orphan effector immunity pairs to be found that are not adjacent to an hcp, (or vgrG or PAAR)?

To clarify the requirement of the chromosomal Hcp1, we revised the text to read: “...mediated the T6SS1-dependent intoxication of *V. natriegens* prey, as indicated by the decline in prey viability when the module was expressed in the parental”

attacker but not when it was expressed in the T6SS1-inactive mutant ($\Delta hcp1$).” We also added a sentence at the end of the relevant Results section to acknowledge that Hcp1b was not sufficient in the absence of Hcp1: “In addition, they suggest that Hcp1b is unable to support T6SS-mediated delivery of the effector in the absence of the canonical T6SS1 Hcp, Hcp1. “

We do not know whether Hcp1b is dispensable. To reach such a conclusion, we will need to investigate the module in the encoding strain and not in the surrogate system, in which the module genes are over-expressed from a plasmid.

Indeed, we and others reported orphan effector and immunity pairs that are not encoded within T6SS gene clusters or next to Hcp, VgrG, or PAAR. This point is mentioned in the Discussion section: “Notably, although this was not directly addressed in this work, additional orphan effectors that are not encoded within T6SS gene clusters or within auxiliary modules next to a hallmark secreted T6SS component have been previously reported in *V. parahaemolyticus* genomes”.

Fig 4. In panel C, it is notable that the expression of Imm4225 with the pEmpty impairs growth. Can a comment be made regarding this result?

On several occasions, we have noticed that immunity proteins that are predicted to localize to the bacterial membrane or to the periplasm are toxic upon over-expression in *E. coli*. We postulate that this is caused by over-crowding the membrane or the Sec system, but we have yet to investigate this phenomenon experimentally. We now mention this observation in the Results section: “Notably, as we previously reported for other immunity proteins that are predicted to function in the bacterial membrane or in the periplasm (37), Imm4225¹⁸⁷⁶⁴ appeared to be mildly toxic when it was over-expressed in *E. coli* (Fig. 4C).”.

Lines 337-40. To construct the "effectorless surrogate T6SS platform" two effectors were deleted, but the Vp1415 is a two aa substitution (lines 177-8), which lacks activity. I am confused, because wouldn't the Vp1415 still be loaded and delivered but not have activity in a neighbor? Has this been shown here or prior? Wouldn't a strain expressing a heterologous effector still experience competition for loading? Clarification here would be useful.

We apologize for the confusion. Indeed, VP1415, which is a specialized effector (and as such it is a structural and necessary part of the spike complex that docks onto a dedicated platform at the tip of the VgrG trimer) is still present in the effectorless system. Our intension was to say that no cargo effectors will compete for loading. This is now clarified in the text: “no endogenous cargo effectors compete with the investigated predicted effector for loading onto the secreted tube and spike...”

Reviewer #3 (Comments for the Author):

The type VI secretion system (T6SS) is mostly present in Gram-negative bacteria and has been used to compete with the host and helps in the invasion process. The Hcp protein bordered with the VgrG/PAAR complex is vital in antibacterial or anti-eukaryotic effectors transportation. The genomic backbone and the functional aspects of T6SS might vary among several bacterial species. In this study, the authors focused on the pan-genome makeup of *Vibrio parahaemolyticus* to better understand the ability of T6SS potential. This study was planned and executed well to address several research questions.

Following are the comments that I would like to set forth.

1. Several abbreviations need to be spelled out in its first use (e.g. PAAR, H-NS, etc.).

The missing abbreviations have been added to the text.

2. Line 120. Mention here the specific virulence function of anti-eukaryotic activity in relation to disease progression.

We now mention the activity, and cite the relevant paper: “T6SS3 is similar to the previously reported T6SS3 of *V. proteolyticus*, which was suggested to have anti-eukaryotic activity and induce inflammasome-mediated cell death in macrophages”.

3. Line 133. The VgrG effector and its alleles are widely reported in potential pathogens like *V. cholerae*, *P. aeruginosa* etc. A comparative account of this effector would be an interesting information.

While many VgrG proteins are known or predicted specialized effectors with a C-terminal toxin domain extension, most VgrGs (including all of the identified VgrG proteins in *V. parahaemolyticus*) are not effectors “per se”, but rather structural components of the T6SS since they lack a C-terminal toxin domain. Therefore, we do not think that this report is the appropriate venue for an overall analysis of VgrG proteins.

4. *V. parahaemolyticus* has a T6SS both on chromosome-1 (T6SS1) and chromosome 2 (T6SS2). Interestingly, the authors have identified two additional T6SSs namely T6SS3 and T6SS4. However, the advantage of having these two new T6SSs to the bacterium has not been discussed in evolutionary or virulence prospective.

We do not offer an evolutionary discussion of the newly identified T6SS3 and T6SS4 because they remain unstudied, and we have no information on the

conditions under which they may be active. Moreover, they do not appear to be restricted to a certain niche (i.e., they are found in isolates from human stool, shrimp hepatopancreas, fish, and environmental settings). The only currently available information that we have is the predicted effectors that they encode, and therefore we hypothesize on their activities in the Discussion section.

5. While discussing cognate immunity protein, the authors should consider the other potential pathogenic vibrios like *V. cholerae*, which occupies the same milieu, while causing the disease.

In the current study, we only tested a few examples of marine pathogens and their ability to resist DUF4225-mediated attack, yet we were still able to observe various resistance phenotypes. Indeed, there may be other Imm4225 homologs or non-immunity protein-mediated defense mechanisms in other pathogens that occupy the same niche as *V. parahaemolyticus*, such as *V. cholerae* that was noted by the reviewer; it will be interesting to investigate their potential defense mechanisms against this widespread toxin in future work.

6. It has been mentioned that two additional T6SSs have been recently acquired by *V. parahaemolyticus*. As the authors well aware that pandemic strains of this pathogen (serovars O3:K6, O4:K68, etc.,) have emerged and spread across the globe causing several outbreaks. It would be interesting to investigate and discuss about these new T6SSs in the pandemic strains. I suggest to screen the pandemic strain WGSs to check the preferential selection of T6SS3 and T6SS4 if any.

As described in the text, the distribution of the newly identified T6SS3 and T6SS4 is quite limited. Following the reviewer's suggestion, we investigated 34 representative genomes of pandemic serotypes, including O3:K6 (Genome IDs: #132, #355, #409, #1163-5, #1182-4, #1450-3, #1455), O4:K12 (#1456-8), O4:K68 (#1454), O4:K8 (#1559) and O4:KUT (#1035-48, #1053). None of these representative genomes contains T6SS3 or T6SS4 (see Supplementary Dataset S2). Moreover, the strains carrying T6SS3 and T6SS4 were collected from diverse niches, as mentioned in our response to the reviewer's comment #4. Therefore, there does not seem to be a correlation between T6SS3/4 and pandemic strains.

September 22, 2022

Dr. Dor Salomon
Tel Aviv University
Clinical Microbiology and Immunology
Tel Aviv
Israel

Re: mSystems00723-22R1 (Multiple T6SSs, mobile auxiliary modules, and effectors revealed in a systematic analysis of the *Vibrio parahaemolyticus* pan-genome)

Dear Dr. Dor Salomon:

Congratulations. Your team's manuscript has been editorially accepted, and I am forwarding it to the ASM Journals Department for publication. For your reference, ASM Journals' address is given below. Before it can be scheduled for publication, your manuscript will be checked by the mSystems production staff to make sure that all elements meet the technical requirements for publication. They will contact you if anything needs to be revised before copyediting and production can begin. Otherwise, you will be notified when your proofs are ready to be viewed.

Publication Fees:

If you would like to submit a potential Featured Image, please email a file and a short legend to mSystems@asmusa.org. Please note that we can only consider images that (i) the authors created or own and (ii) have not been previously published. By submitting, you agree that the image can be used under the same terms as the published article. File requirements: square dimensions (4" x 4"), 300 dpi resolution, RGB colorspace, TIF file format.

We recognize that the video files can become quite large, and so to avoid quality loss ASM suggests sending the video file via <https://www.wetransfer.com/>. When you have a final version of the video and the still ready to share, please send it to mSystems staff at mSystems@asmusa.org.

Sincerely,

Seth Bordenstein
Editor, mSystems

Journals Department
Dataset S1: Accept
Dataset S2: Accept
Dataset S3: Accept
Fig. S2: Accept
Movie S1: Accept
Fig. S1: Accept
Fig. S4: Accept
Fig. S3: Accept
Table S1: Accept
Dataset S4: Accept